# Phytoplasma-induced alterations in endophytic bacterial communities in *Paulownia*: implications for witches' broom

Xuefei Tang,[1] Tuoyan Chen,[2] Xiaoqiao Zhai,[3] Jing Huang,[1] Yifei Sun,[2] Yuchen Yang,[2] Zhenli Zhao,[1] Yanshuo Pan,[1] Yuhao Huang,[2] Xuanzhen Li,[1] Guoqiang Fan[1]

**ABSTRACT** Paulownia witches' broom (PaWB), caused by phytoplasma, threatens global *Paulownia* cultivation. Although phytoplasma is known to manipulate host physiology, their influence on the plant-associated microbiome, particularly at the tissue-specific level, remains unclear. Here, we integrated scanning electron microscopy, nested PCR, and 16S rRNA sequencing to investigate the morphological characteristics, phytoplasma infection, and bacterial communities across four compartments (leaves, branches, roots, and rhizosphere) in healthy, symptomatic, and asymptomatic *P. fortunei* trees. We found that PaWB induces pronounced external morphological abnormalities without significantly altering internal tissue structures. Notably, phytoplasma was detected not only in symptomatic tissues but also at low abundance in asymptomatic compartments, indicating a latent infection state. Phytoplasma proliferation in symptomatic leaves and branches was accompanied by a dramatic loss of bacterial diversity and a collapse in microbial interaction network complexity, while the root and rhizosphere microbiota remained comparatively stable. Random forest modeling identified *Candidatus* phytoplasma and ten other bacterial genera as key predictors of PaWB status. Microbial function predictions further revealed that disrupted carbohydrate degradation and tryptophan metabolism in diseased tissues may promote the expansion of opportunistic microbes, potentially exacerbating PaWB symptom development. Our study provides the first compelling evidence that phytoplasma infection drives tissue-specific microbiome collapse in *Paulownia*, disrupting microbial networks and reprogramming functional pathways well before visible symptoms emerge. These previously unrecognized microbial and metabolic signatures represent robust early-warning biomarkers and offer strategic targets for precision diagnostics and disease intervention, marking a significant advance in our understanding and management of PaWB.

**IMPORTANCE** Paulownia witches' broom (PaWB) disease poses a severe threat to global *Paulownia* cultivation, yet its microbiome-related mechanisms remain poorly understood. Here, we show that phytoplasma infection alters the external *P. fortunei* morphology and disrupts the composition, structure, and function of endophytic bacterial communities in aerial tissues. These microbial shifts are closely linked to symptom development, with latent infections also detected in asymptomatic tissues. Eleven microbial markers, including *Candidatus* Phytoplasma, enable accurate disease prediction. Predictions of functional shifts in carbohydrate and tryptophan metabolism further implicate microbiome alterations in symptom development. Our study contributes to a deeper understanding of the complex interaction mechanism among *Paulownia*, microorganisms and phytoplasma during the occurrence of PaWB and offer a theoretical foundation for sustainable management of PaWB.

**KEYWORDS** Paulownia witches' broom, phytoplasma, endophytic bacteria, 16S rRNA sequencing, microbial network

**Peer Reviewer** Franco Daniel Franco, Instituto de Patologia Vegetal CIAP-INTA, Córdoba, Argentina

Address correspondence to Xuanzhen Li, xzli@henau.edu.cn, or Guoqiang Fan, zlxx64@126.com.

Xuefei Tang and Tuoyan Chen contributed equally to this article. The author order was determined based on their contribution to the article.

The authors declare no conflict of interest.

Plants host diverse microbial communities across their tissues, including leaves, stems, roots, and rhizosphere. These microbiomes profoundly influence the plant productivity and health by modulating nutrient uptake, hormone biosynthesis, and defense responses against pathogens (1–3). In turn, plants actively shape their associated microbiota, especially under biotic stress, by recruiting beneficial microbes that contribute to systemic resistance (4–7). For instance, *Pseudomonas* sp. CMR12a produces cyclic lipopeptides that trigger systemic resistance in rice and bean against pathogens (8). Similar microbiome-mediated defense mechanisms have been observed in sugar beet (4), capsicum (5), wheat (6), *Amorphophallus* (9), and *Arabidopsis thaliana* (7). Therefore, regulating plant microbiomes is increasingly recognized as a sustainable strategy for disease control and yield improvement in agriculture and forestry. A deep understanding of how plants modulate these communities and their functions under pathogen stress is essential for improving plant health and maximizing productivity (10).

*Paulownia*, a fast-growing deciduous treegenus native to China, is an important timber and shelterbelt tree species. It is widely valued for its timber, carbon sequestration capacity, and ecological and biomedical applications (11, 12). However, its productivity and survival are severely threatened by Paulownia witches' broom disease (PaWB), a destructive condition caused by phytoplasma infection and often dubbed the "cancer" of *Paulownia*. PaWB leads to abnormal shoot proliferation, shortened internodes, and eventual plant death, with disease incidence exceeding 80% in severely affected regions, resulting in annual economic losses of billions of yuan in China alone (13–15). Effective management of PaWB remains elusive due to the unculturable nature of phytoplasma. Recent findings suggest that phytoplasma effectors, such as SAP54, may trigger symptom development by interfering with host hormone metabolism (16). Despite growing insights into host responses to the infection at genomic (16, 17), transcriptomic (18–21), proteomic (22, 23), metabolic (22), and physiological levels (24, 25), the impacts of PaWB pathogenesis on the microorganisms in *Paulownia* remain largely unexplored. In particular, it is unclear how phytoplasma affects the composition and function of microbial communities across *Paulownia* compartments. Whether phytoplasma infection induces compartment-specific disruption of endophytic bacterial networks and potential early-warning signals of disease onset remains an open question.

To elucidate how phytoplasma infection influences host phenotypes and associated microbiota, we investigated tissue morphology, phytoplasma colonization, and microbial community patterns across four compartments (leaves, branches, roots, and rhizosphere) in healthy, symptomatic, and asymptomatic *P. fortunei* trees. We employed scanning electron microscopy, nested PCR, and 16S rRNA sequencing, complemented by microbial network analysis, random forest modeling, and functional prediction. Our objectives were to (i) assess how phytoplasma infection alters microbiome diversity and assembly processes across plant compartments; (ii) determine whether these alterations are compartment-specific; (iii) identify potential microbial biomarkers and functional pathways associated with PaWB symptom development.

## MATERIALS AND METHODS

### Field trial

A field trial was conducted in Yuzhou City, Henan Province, China—an area ideal for *Paulownia* growth. Established in 2017 on a 1-hectare former wheat field, *P. fortunei* was planted at 4 m (within rows) and 5 m (between rows) spacing. At sampling, trees averaged 5–6 m in height, with approximately 10% showing WB symptoms.

### Morphological observation of the P. fortunei samples using scanning electron microscopy

For scanning electron microscopy (SEM), fresh branch segments of *P. fortunei* were cut into ~2-mm-thick transverse sections. The sections were affixed to aluminum stubs using

conductive carbon tape and sputter-coated with gold (35 s, Cressington 108Auto). A field-emission SEM (FEI Q45) was used to observe the specimens at an accelerating voltage of 5 kV. Overview images were captured at 52× magnification, followed by detailed imaging of selected regions at 150×.

### Phytoplasma detection in P. fortunei via nested PCR

Total genomic DNA was extracted from stem and root samples using a Plant Genomic DNA Kit (TIANGEN, Beijing, China) according to the manufacturer's protocol. DNA integrity and concentration were verified by 1% agarose gel electrophoresis and NanoDrop 2000 spectrophotometry, respectively.

Nested PCR was performed in 25 µL reactions to amplify the phytoplasma 16S rRNA gene. The primary reaction, using primers R16mF1/R16mR1, was templated with 2 µL of genomic DNA. For the nested reaction, 1 µL of a 1:20 diluted primary PCR product served as the template with primers R16F2/R16R2. The thermal cycling profile included an initial denaturation at 94°C for 3 min, followed by 30 amplification cycles and a final extension at 72°C for 10 min. For complete reaction components and detailed cycling parameters, see Table S1. All primer sequences are listed in Table S2. PCR products were visualized on 1.0% agarose gels in 1× TAE buffer under UV light.

### Sample collection and processing

A total of six *P. fortunei* trees (three healthy and three symptomatic) were selected for microbiome analysis (Fig. 1; Table 1). Four sample types were collected from each tree: pooled leaves (10 per branch), branch segments (8 × 1 cm pieces), roots (from 5 to 10 cm depth), and rhizosphere soil. From each diseased tree, both a symptomatic and an asymptomatic (the absence of visible PaWB symptoms in diseased *Paulownia*) branch were sampled. Leaf, branch, and root tissues were surface-sterilized via sequential washes in sterile water (30 s), 70% (vol/vol) ethanol (1 min), 2.0% sodium hypochlorite (3 min), and a final rinse in sterile water (30 s). All samples were immediately stored at −80°C pending DNA extraction.

### DNA extraction, library preparation, and Illumina sequencing

Total genomic DNA was extracted from tree and soil samples using the E.Z.N.A. Soil DNA Kit (Omega Bio-Tek, USA) according to the manufacturer's protocol. The integrity of the extracted DNA was verified on 1% agarose gels, while the concentration and purity were measured with a NanoDrop 2000 spectrophotometer (Thermo Fisher Scientific, USA).

To analyze the bacterial communities, the V5–V7 hypervariable regions of the 16S rRNA gene were amplified using the primer pair 799F/1193R, which minimizes

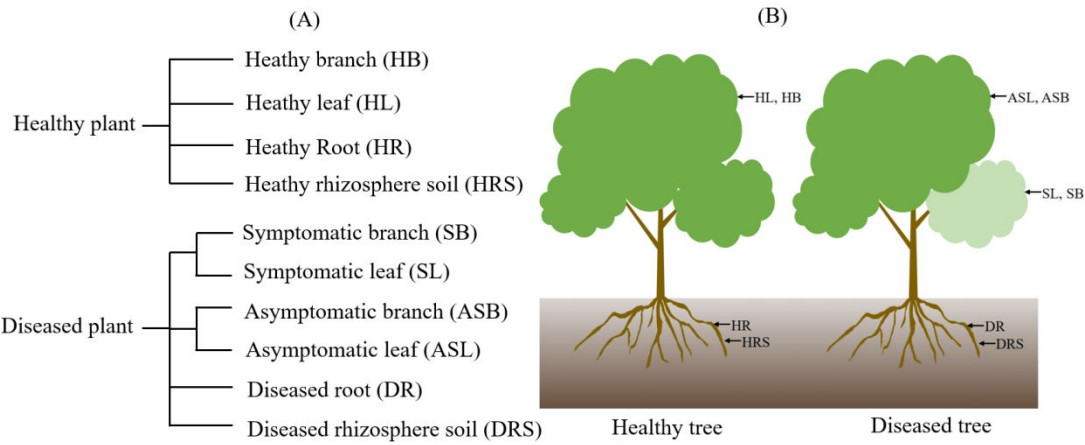

**FIG 1** (A and B) Sources of tested samples and their abbreviations. HB: healthy branch; HL: healthy leaf; HR: healthy root; HRS: healthy rhizosphere soil; SB: symptomatic branch; SL: symptomatic leaf; ASB: asymptomatic branch; ASL: asymptomatic leaf; DR: diseased root; DRS: diseased rhizosphere soil.

**TABLE 1** Sample information

| Sample name | Compartment | Health condition | Sample source | Duplicates |
|---|---|---|---|---|
| HL | Leaf | Healthy | Healthy plant | 6 |
| ASL | Leaf | Asymptomatic | Diseased plant | 6 |
| SL | Leaf | Symptomatic | Diseased plant | 6 |
| HB | Branch | Healthy | Healthy plant | 6 |
| ASB | Branch | Asymptomatic | Diseased plant | 6 |
| SB | Branch | Symptomatic | Diseased plant | 6 |
| HR | Root | | Healthy plant | 6 |
| DR | Root | | Diseased plant | 6 |

amplification of host plant mitochondrial and chloroplast DNA. All PCRs were performed in triplicate. The specific components of the 20 µL reaction mixture and the thermal cycling program, run on an ABI GeneAmp 9700 thermocycler, are detailed in Table S3. Primer sequences are listed in Table S4.

Following amplification, the triplicate products were pooled and purified from a 2% agarose gel using the AxyPrep DNA Gel Extraction Kit (Axygen Biosciences, USA). The purified amplicons were quantified with a Quantus Fluorometer (Promega, USA) and pooled in equimolar concentrations to create the final library. Paired-end (2 × 300 bp) sequencing was performed on the Illumina MiSeq platform by Majorbio Bio-Pharm Technology Co. Ltd. (Shanghai, China). The raw sequencing reads have been deposited in the NCBI Sequence Read Archive (SRA) under accession number PRJNA821669.

## Microbial community analysis

Raw 16S rRNA reads were processed in a pipeline using fastp (v0.20.0) (26) for quality filtering and demultiplexing, followed by FLASH (v1.2.7) (27) to merge paired-end reads. Operational taxonomic units (OTUs) were then clustered from the merged, high-quality sequences at 97% similarity using UPARSE (v7.1) (28), which also removed chimeras. Finally, taxonomy was assigned using the RDP Classifier (v2.2) (29) with a confidence threshold of 0.7. Specific parameters for quality control, read merging, and OTU clustering are detailed in Table S5.

Ecological and statistical analyses were performed in R (30). Alpha diversity metrics (Shannon, Chao1, and Pielou's evenness) and beta diversity, visualized by principal coordinate analysis (PCoA) (31), were calculated using the vegan package. The effects of sampling site and disease status on community structure were tested with PERMANOVA (adonis2 function) (32). Bacterial co-occurrence networks were constructed with WGCNA (33), where edges representing significant associations (FDR-adjusted $P < 0.001$; correlation ≥ 0.78) (34) were retained for analysis with the igraph package. Network topology was analyzed and visualized using the igraph (35) package. The randomForest package (36) was used to identify biomarker species. All statistical comparisons between groups were performed using one-way ANOVA with an LSD *post hoc* test (agricolae package).

The functional potential of the bacterial communities was predicted from 16S rRNA data using PICRUSt2 (37). This process involved placing OTU representative sequences into a reference phylogeny to infer gene family abundances, which were then mapped to the MetaCyc database (38) to predict metabolic pathways. To identify key functional differences between sample groups (e.g., healthy vs diseased), the predicted pathway abundances were analyzed using a random forest classifier (39) to determine feature importance, followed by differential abundance testing with one-way ANOVA and Benjamini-Hochberg FDR correction (34) for statistical validation.

## RESULTS

### Phenotypic comparison of the asymptomatic and symptomatic *Paulownia* branches

Healthy and asymptomatic branches showed minimal morphological changes—thick, robust twigs with no visible abnormalities—whereas diseased branches exhibited increased secondary shoot proliferation with notably thinner shoots, a known PaWB characteristic. Even after approximately 3 years of PaWB infection, symptoms remained confined to individual branches, suggesting potential immunity or resistance in *Paulownia*. (Fig. 2)

Scanning electron microscopy of transverse stem sections revealed no significant differences in internal tissue architecture among HB, ASB, and SB samples (Fig. 3). All stems consistently showed a hollow pith and numerous small perforations in the xylem.

### Nested PCR detection of phytoplasma in healthy and diseased *Paulownia*

Nested PCR was performed to detect phytoplasma in stem and root tissues. The expected ~1,200 bp amplicon was detected in SB and ASB samples, as well as in DR samples (Fig. 4A and B). No amplification was observed in HB or HR control samples. Notably, the PCR product from asymptomatic stems was consistently fainter than that from symptomatic stems (Fig. 4A).

### α-Diversity of bacterial communities in different tree compartments

Using 16S rRNA gene sequencing, we assessed the α diversity of bacterial communities in various *Paulownia* compartments (Fig. 5). We found that symptomatic leaves had significantly lower Shannon-Wiener, Chao1, and Pielou's evenness indices than healthy and asymptomatic leaves (Fig. 5A). A similar reduction was observed in symptomatic branches compared to their healthy and asymptomatic counterparts (Fig. 5B).

In asymptomatic leaves, the Shannon-Wiener and Pielou's indices were higher than in healthy leaves, although the Chao1 index was lower (Fig. 5A). No significant differences were found between healthy and asymptomatic branches nor between diseased and healthy roots or rhizosphere soil (Fig. 5B and D). The effect of WB on bacterial α-diversity was only seen in the above-ground parts (leaves and branches), not the below-ground parts (roots and soil).

### PCoA of bacterial communities in different compartments

PCoA using Bray-Curtis distances revealed that for the endophyte community, PCoA1, PCoA2, and PCoA3 explained 32.36%, 16.53%, and 11.92% of the variation, respectively (Fig. 6A). Healthy and asymptomatic branch samples clustered together, distinct from symptomatic branches. Similarly, healthy and asymptomatic leaf samples—though partially overlapping—were notably different from symptomatic leaves. These findings suggest that WB was the primary driver of variation in the bacterial communities within the leaf and branch compartments.

(A)                                  (B)                                  (C)

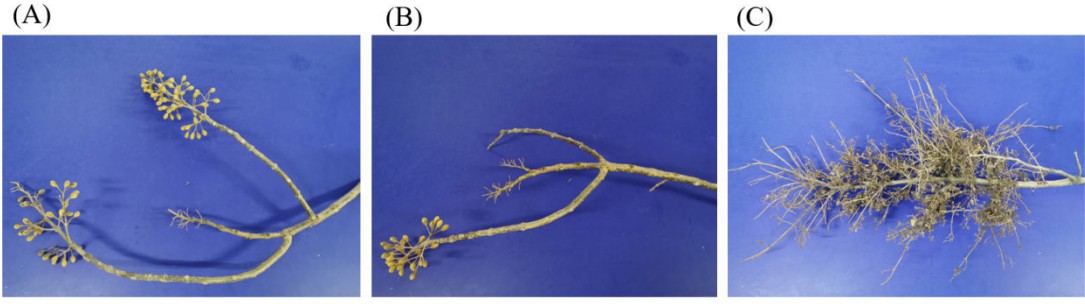

**FIG 2** The appearance of symptomatic and asymptomatic branches. (A) Healthy; (B) asymptomatic; (C) symptomatic.

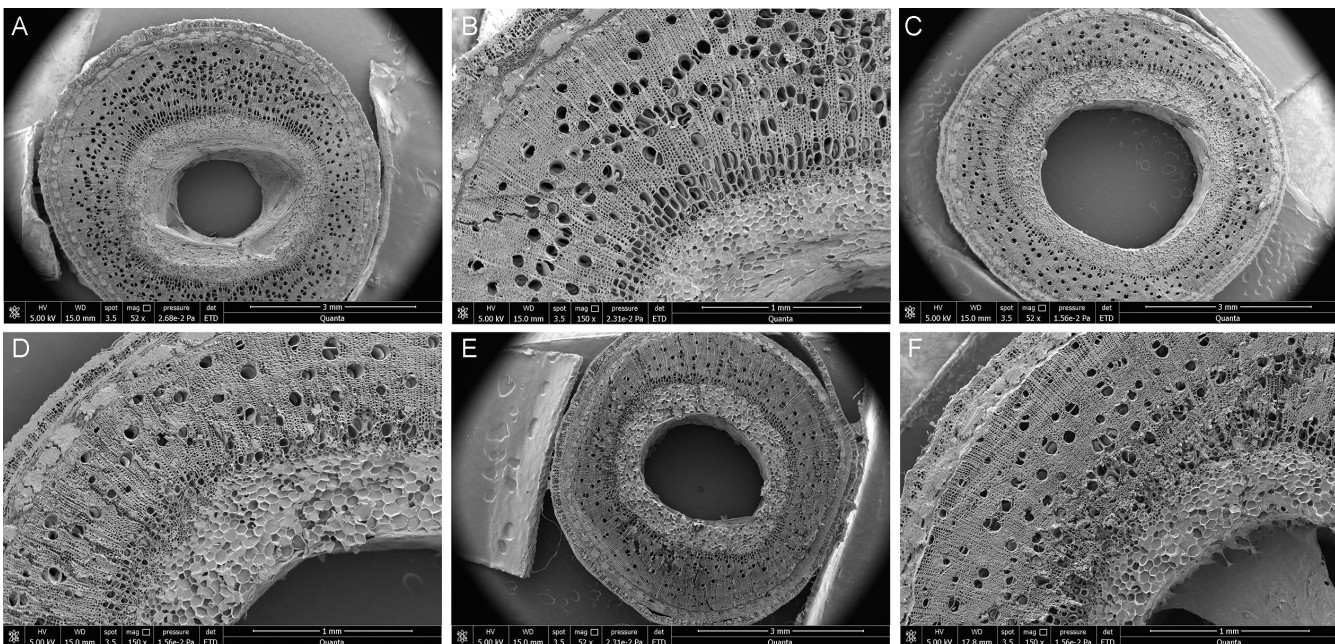

**FIG 3** Microstructure of transverse sections of healthy and diseased *Paulownia* stems. (A and B) Healthy stems; (C and D) asymptomatic stems from diseased plants; (E and F) symptomatic stems from diseased plants.

Moreover, healthy and diseased root samples formed a distinct cluster separate from leaves and branches (Fig. 6A), and diseased versus healthy rhizosphere soils were clearly separated (Fig. 6B). PERMANOVA confirmed that both plant compartments ($R^2 = 0.22$, $P < 0.001$) and health status ($R^2 = 0.18$, $P < 0.001$) significantly influence bacterial community composition, with compartments having the greatest impact (Table 2).

## Random forest models pinpoint *Candidatus* Phytoplasma as the core predictive biomarker

Prediction accuracy was highest at the phylum level (100%), followed by the class and genus levels (both 98.33%), with total reads reaching 11608 (Fig. 7A and E), under a five-repetition, tenfold cross-validation. Ultimately, the genus level was selected for WB prediction, where a model using 11 key genera (Fig. 7B), namely, *Candidatus* Phytoplasma, *Pseudomonas*, *Sphingomonas*, *Novosphingobium*, *Bacillus*, *Microbacterium*, unclassified Xanthomonadaceae, unclassified Oxalobacteraceae, *Rhodococcus*, unclassified Comamonadaceae, and *Bdellovibrio,* was employed. Among these, *Candidatus* Phytoplasma had the greatest impact, followed by *Novosphingobium and Sphingomonas* (Fig. 7C). Notably, we found a clear gradient in the abundance of *Candidatus* Phytoplasma that corresponded with the trees' health status (Fig. S1). In trees with disease symptoms, phytoplasma levels were exceptionally high in leaves (98.8%) and branches (86.4%), with lower levels in roots (17.7%). The abundance decreased sharply in asymptomatic parts of diseased trees (1.7% in leaves and 1.0% in branches) and was further reduced to trace levels in healthy trees (0.1% in leaves, 0.2% in branches, and 2.4% in roots). However, phytoplasmas were undetectable in rhizosphere soil. This quantitative gradient strongly indicates that the development of WB symptoms is closely linked to a high titer of phytoplasma, particularly within the leaves and branches.

## Phytoplasma infection has a tissue-specific impact on bacterial community structure and stability

The bacterial communities in symptomatic leaves and branches were significantly different from those in healthy or asymptomatic tissues. In healthy and asymptomatic

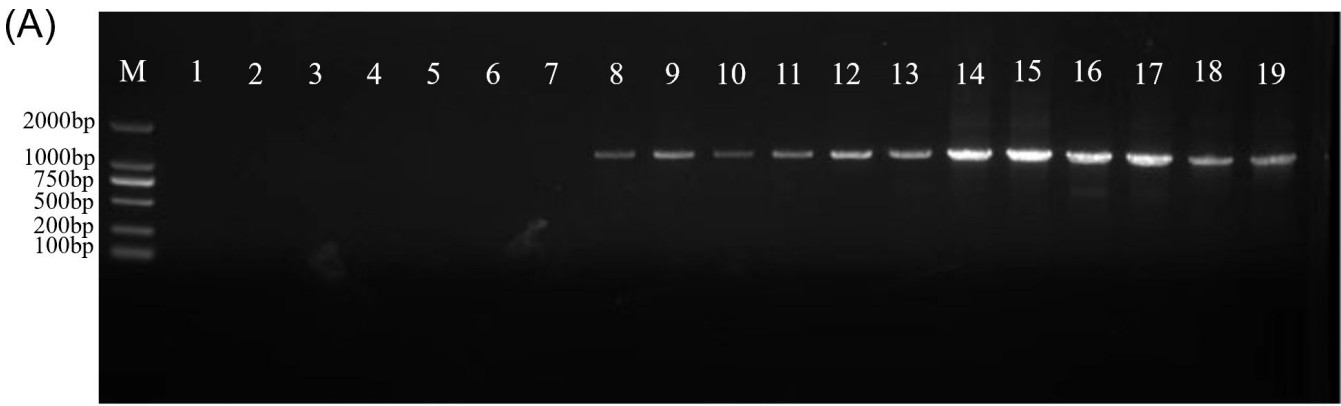

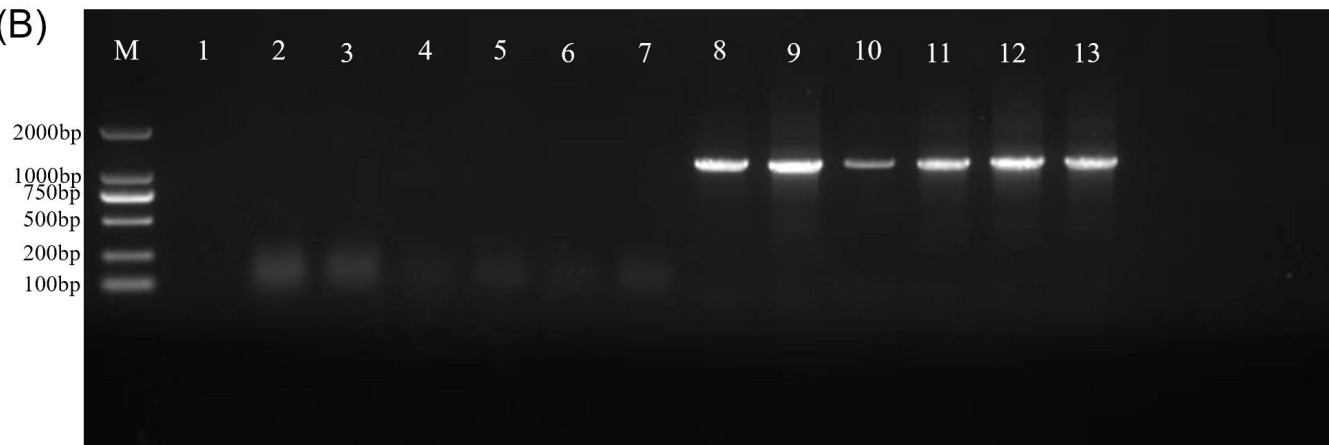

**FIG 4** (A) Detection of phytoplasma in branches of healthy, diseased, and asymptomatic *Paulownia*; (B) detection of phytoplasma in healthy and diseased *Paulownia*. (A) Lanes: M, 2,000 bp marker; 1, ddH₂O; 2–7, HB; 8–13, ASB; 14–19, SB. (B) Lanes: M, 2,000 bp marker; 1, ddH₂O; 2–7, HR; 8–13, DR.

samples across all tissues, Proteobacteria was the dominant phylum (44.7%–78.4%). However, the dominance of Firmicutes (87.6% and 99.1%, respectively), primarily *Candidatus* phytoplasma, in symptomatic leaves and branches (Fig. S2; Fig. 7D), reflects pathogen expansion, rendering phylum-level analysis of Firmicutes uninformative. Subsequent analyses therefore focused on genus-level resolution.

In contrast, the belowground microbial communities were largely unaffected by the disease. In both roots and rhizosphere soils, the community structure remained similar across healthy and diseased trees. These compartments were consistently dominated by Proteobacteria, Actinobacteriota, and Acidobacteriota (Fig. 7D). Statistical analyses confirm that the pathogen disrupts the microbial community's diversity and stability in aerial tissues. First, the abundance of *Candidatus* Phytoplasma in leaves and branches was negatively correlated with all three α-diversity indices ($P < 0.05$), while other bacteria generally showed a positive correlation. This loss of diversity directly translated to a collapse in community stability.

Co-occurrence network analysis showed that symptomatic samples had significantly fewer nodes and edges, indicating lower network complexity (Fig. S4 and S5). In fact, α-diversity was strongly and positively correlated with network complexity in leaves and branches (e.g., $R^2 = 0.83$–$0.84$, $P < 0.001$) (Fig. S5). Consequently, *Candidatus* phytoplasma was also negatively correlated with the number of network nodes and edges ($P < 0.01$), further demonstrating its role in reducing community stability (Fig. S8). In sharp contrast, all these correlations were weak or nonsignificant in the roots and rhizosphere soil,

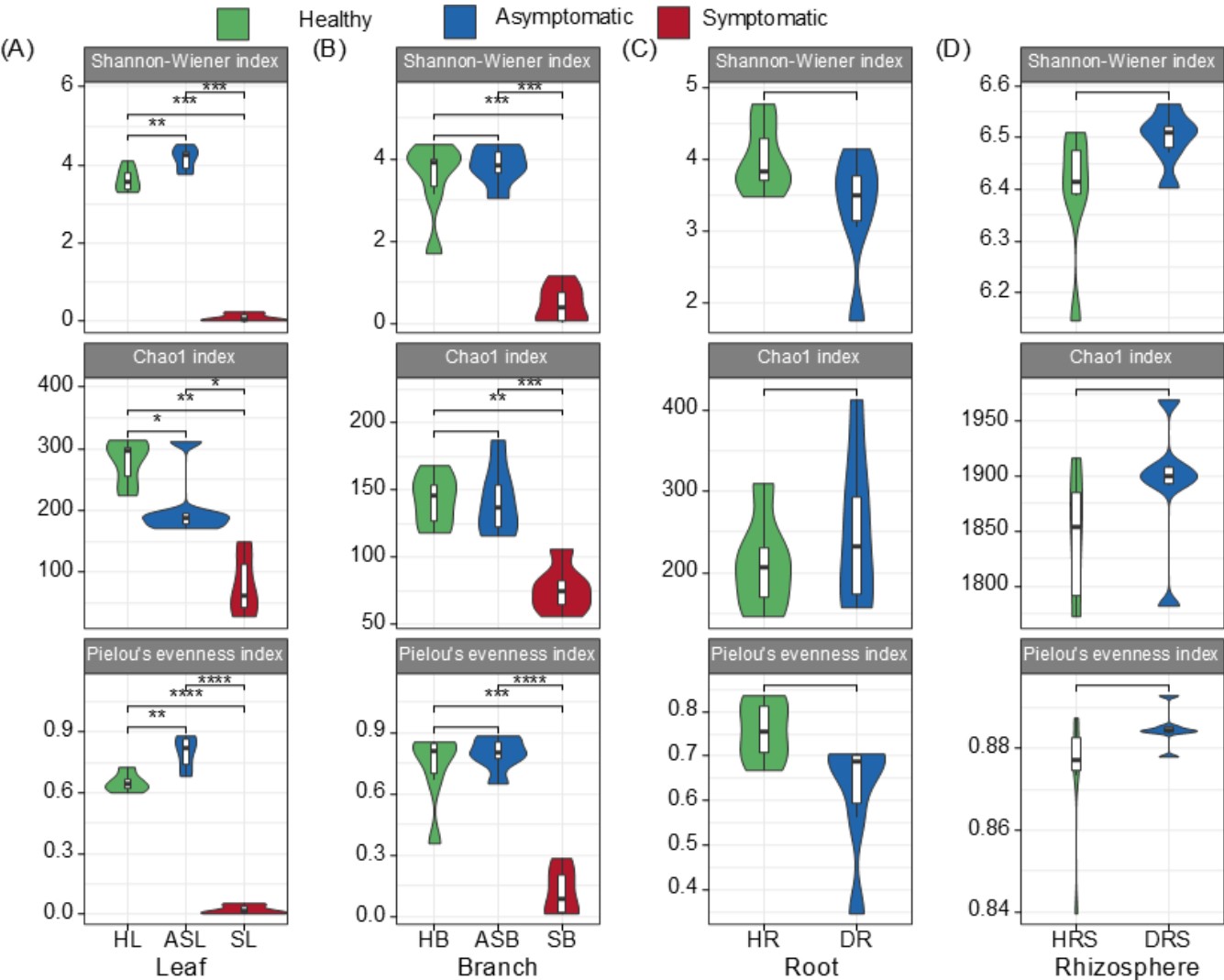

**FIG 5** (A–D) α-Diversity of bacterial communities in different samples. Significance: * represents $P < 0.05$; ** represents $P < 0.01$; *** represents $P < 0.001$. HB: healthy branch; HL: healthy leaf; HR: healthy root; HRS: healthy rhizosphere soil; SB: symptomatic branch; SL: symptomatic leaf; ASB: asymptomatic branch; ASL: asymptomatic leaf; DR: diseased root; DRS: diseased rhizosphere soil.

underscoring the limited impact of the pathogen on belowground microbial communities (Fig. S3, S6, and S8).

## Predictions of functional roles of microbiomes in different compartments under varying conditions

PICRUSt2 predictions (Fig. S7) revealed that each compartment exhibited significant enrichment of a distinct function: symptomatic leaves of carbohydrate degradation; asymptomatic leaves of amino-acid biosynthesis; healthy leaves of cell-wall biosynthesis; diseased roots of gallate degradation II; healthy roots of aromatic-compound degradation (e.g., catechol); symptomatic branches of 2-aminophenol degradation; and healthy branches of L-tryptophan degradation via the *Geobacillus* pathway.

## Endophytic network responses

Our co-occurrence networks (Fig. 8; Table S5) show that SL and SB samples exhibit reduced average degree and a lower proportion of negative edges compared with their healthy counterparts. In ASL and ASB samples, the average degree remains

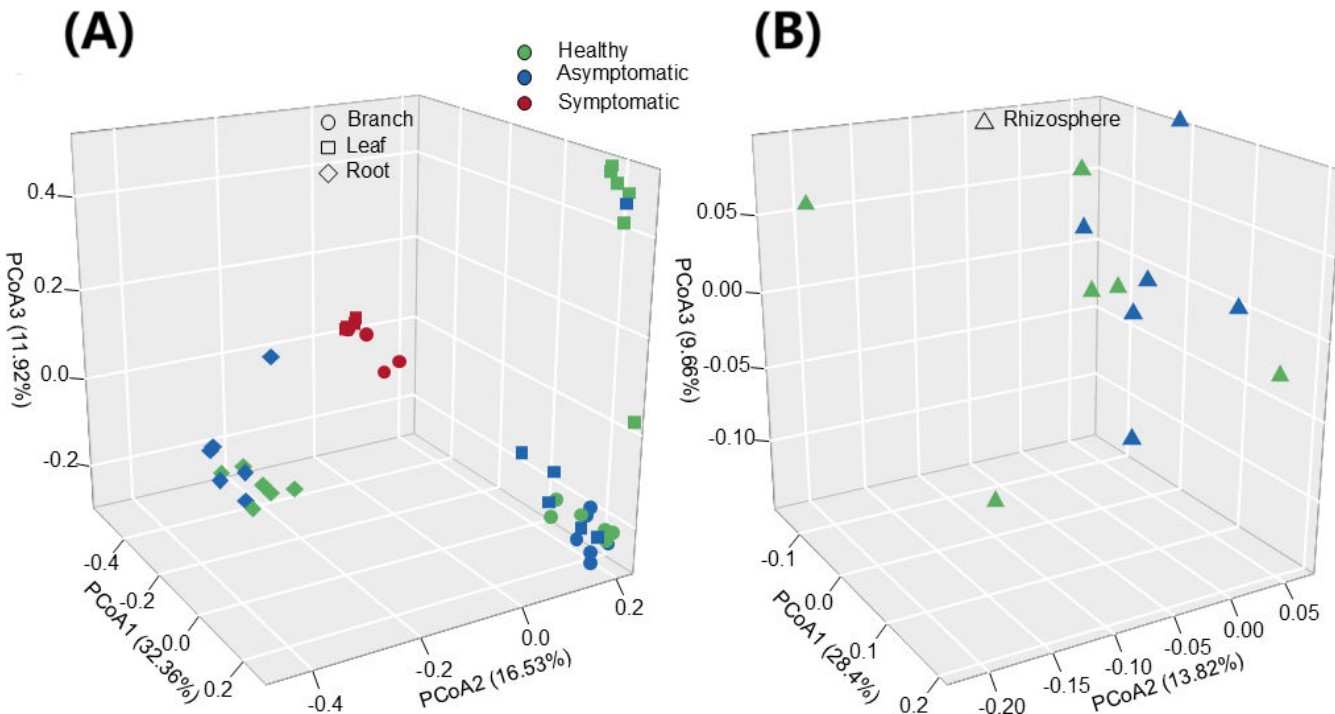

**FIG 6** Principal coordinate analysis of bacterial communities. (A) Compartments of *Paulownia*; (B) soil of the rhizosphere

essentially unchanged, while the fraction of negative edges increases. DR shows a higher average degree but a slightly reduced negative-edge proportion relative to HR. Similarly, diseased rhizosphere soil networks display decreases in both average degree and negative-edge proportion compared with healthy rhizosphere soil. These shifts demonstrate that witches' broom disease systematically reshapes bacterial co-occurrence topology across both endophytic and rhizospheric communities.

## DISCUSSION

Our study not only documents pathogen-induced diversity loss but also uncovers a novel, hierarchical disruption: phytoplasma first collapses microbial interaction networks in aerial tissues (Fig. S4 to S6), which then leads to α-diversity reduction and symptom emergence. This tissue-specific network destabilization provides a mechanistic explanation for the "latency-to-symptom" transition (Fig. 9), which was previously attributed solely to pathogen titer (40).

### Phytoplasma infection drives tissue-specific microbiome collapse that precedes visible symptom expression

PaWB, caused by phytoplasma infection, has long been recognized for its devastating effects on the tree architecture and productivity, yet its microbiome-level indicators have remained poorly understood. Our study fills a critical gap by providing the first direct evidence that phytoplasma disrupts the *Paulownia* endophytic microbiome in a tissue-specific and symptom-correlated manner, fundamentally reshaping bacterial

**TABLE 2** PERMANOVA results for habitat and symptoms on bacterial communities[a]

| Parameter | df | $R^2$ | F value | P |
|---|---|---|---|---|
| Compartments | 3 | 0.22 | 11.26 | <0.001 |
| Symptoms | 2 | 0.18 | 13.84 | <0.001 |
| Compartments × symptoms | 4 | 0.29 | 11.39 | <0.001 |

[a]df is the degree of freedom; $R^2$ is the goodness of fit; P is the significance.

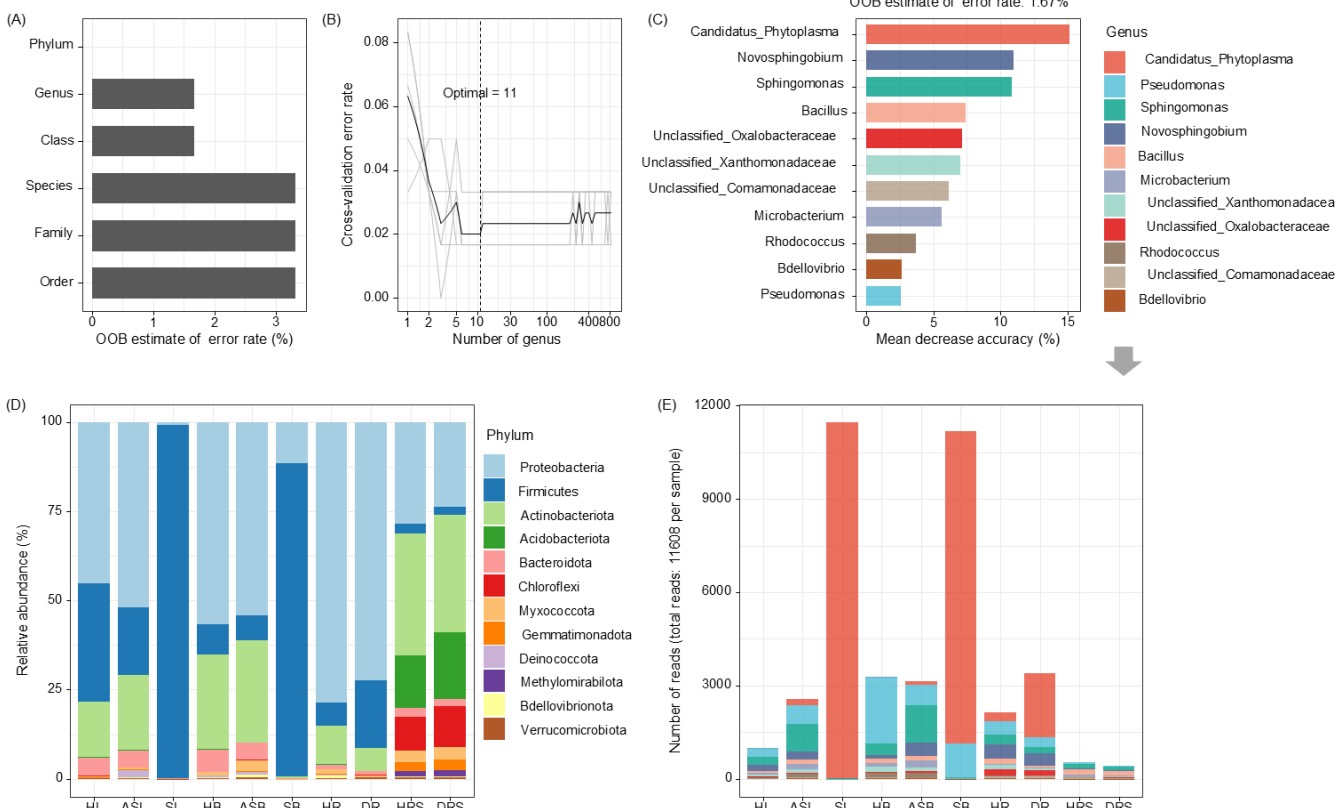

**FIG 7** Random forest analysis. (A) Top discriminant taxa ranked by mean decrease accuracy; (B) out-of-bag error rate across models, highlighting an optimal set of 11 genera; (C) mean decrease accuracy of these 11 genera, with *Candidatus* Phytoplasma as the top predictor; (D) composition of the bacterial community in the phylum level in the different compartments of *Paulownia*; (E) key genera distributions among compartments. HB: healthy branch; HL: healthy leaf; HR: healthy root; HRS: healthy rhizosphere soil; SB: symptomatic branch; SL: symptomatic leaf; ASB: asymptomatic branch; ASL: asymptomatic leaf; DR: diseased root; DRS: diseased rhizosphere soil.

community structure, interaction networks, and functional profiles—often well before external symptoms become visible.

## Latent colonization and aboveground microbiome destabilization underpin symptom emergence

Our results demonstrate that phytoplasma not only accumulates massively in symptomatic leaves and branches (up to 98.8% relative abundance) but is also detectable at trace levels in asymptomatic tissues, suggesting a latent infection state. Importantly, phytoplasma abundance exhibited a strong negative correlation with bacterial α-diversity, underscoring that microbiome collapse is an early, mechanistically significant marker of disease progression. While this pattern was consistently observed in aerial compartments, the belowground microbiota (roots and rhizosphere) remained structurally stable and resilient, highlighting a clear tissue-specific vulnerability.

Based on our bacterial co-occurrence networks (Fig. 8; Table S5), we concluded that the outbreak of PaWB follows two distinct scenarios (Fig. 9) in the above-ground tissues (leaves and branches), determined by the response of the endophytic community to phytoplasma invasion.

In the first scenario, community resilience prevents disease. This is observed in ASL and ASB, where network complexity remains stable while antagonistic interactions (negative edges) increase, indicating active pathogen suppression.

In the second scenario, community collapse triggers the disease. This occurs in SL and SB, where the network shows significantly reduced complexity and interactions. This loss

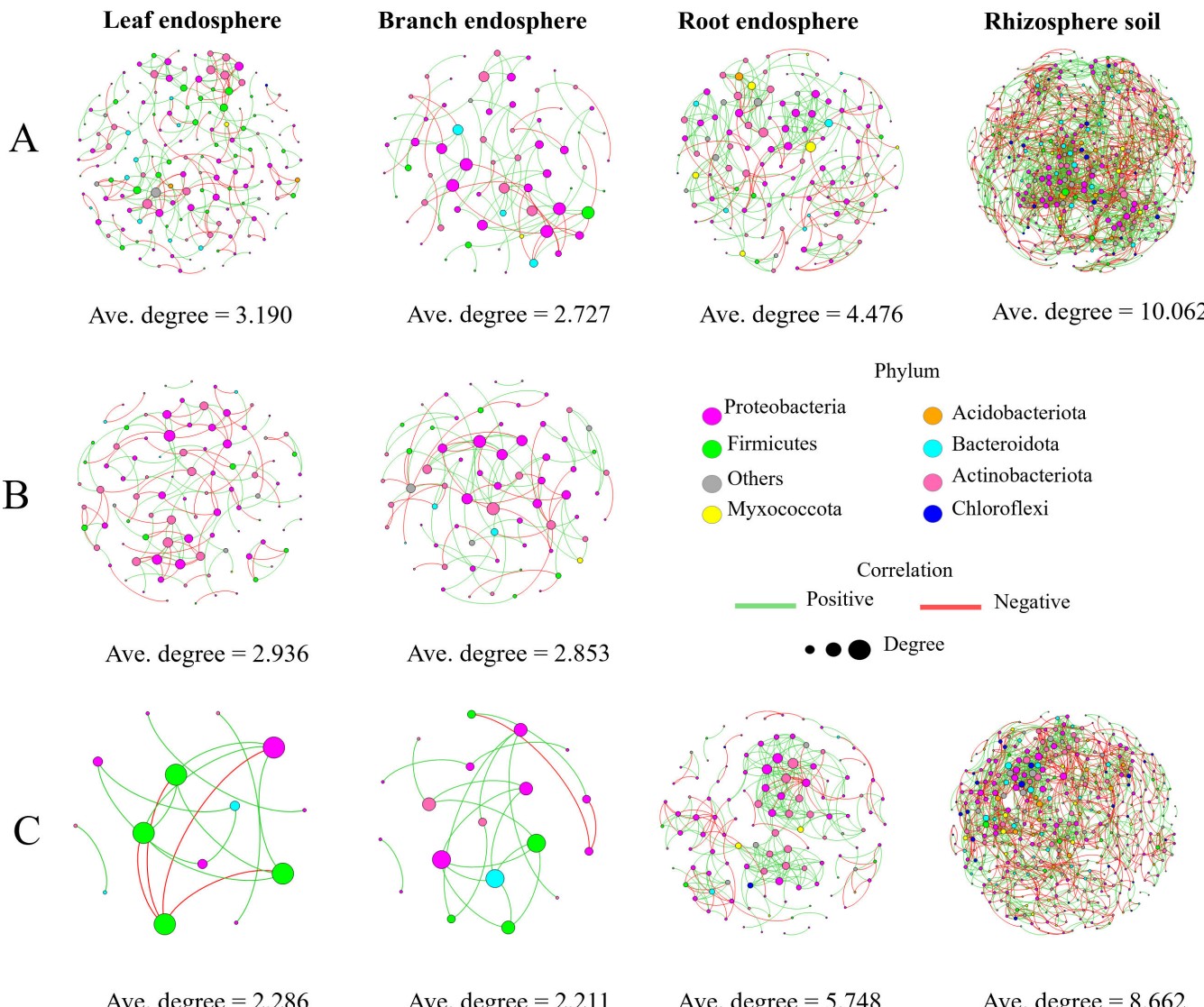

**FIG 8** Bacterial co-occurrence networks in different compartments of *Paulownia*. (A) Bacterial networks in leaf, branch, root, and rhizosphere soil compartments in healthy plants; (B) bacterial networks in asymptomatic leaf and branch compartments in diseased plants; (C) bacterial networks in symptomatic leaf and branch, as well as root and rhizosphere soil compartments in diseased plant).

of stability leads to symptom development and likely initiates a vicious cycle of pathogen proliferation.

The response in the roots, however, was markedly different. Unlike the above-ground parts, the bacterial network in DR did not collapse; instead, its complexity (average degree) became even higher than in HR. This robust, and even enhanced, community network structure is likely the primary reason why the roots can resist the impact of phytoplasma and remain symptom-free.

These findings advance a conceptual model in which microbial community stability in asymptomatic tissues buffers against pathogen proliferation and symptom onset. Once a critical threshold of phytoplasma load is reached, however, community destabilization and loss of ecological resilience may initiate a self-reinforcing disease cycle. This perspective not only explains the observed spatial restriction of PaWB symptoms but also reconciles latent pathogen detection with inconsistent symptom development—a pattern noted in other systems, such as eucalyptus and chestnut (41, 42).

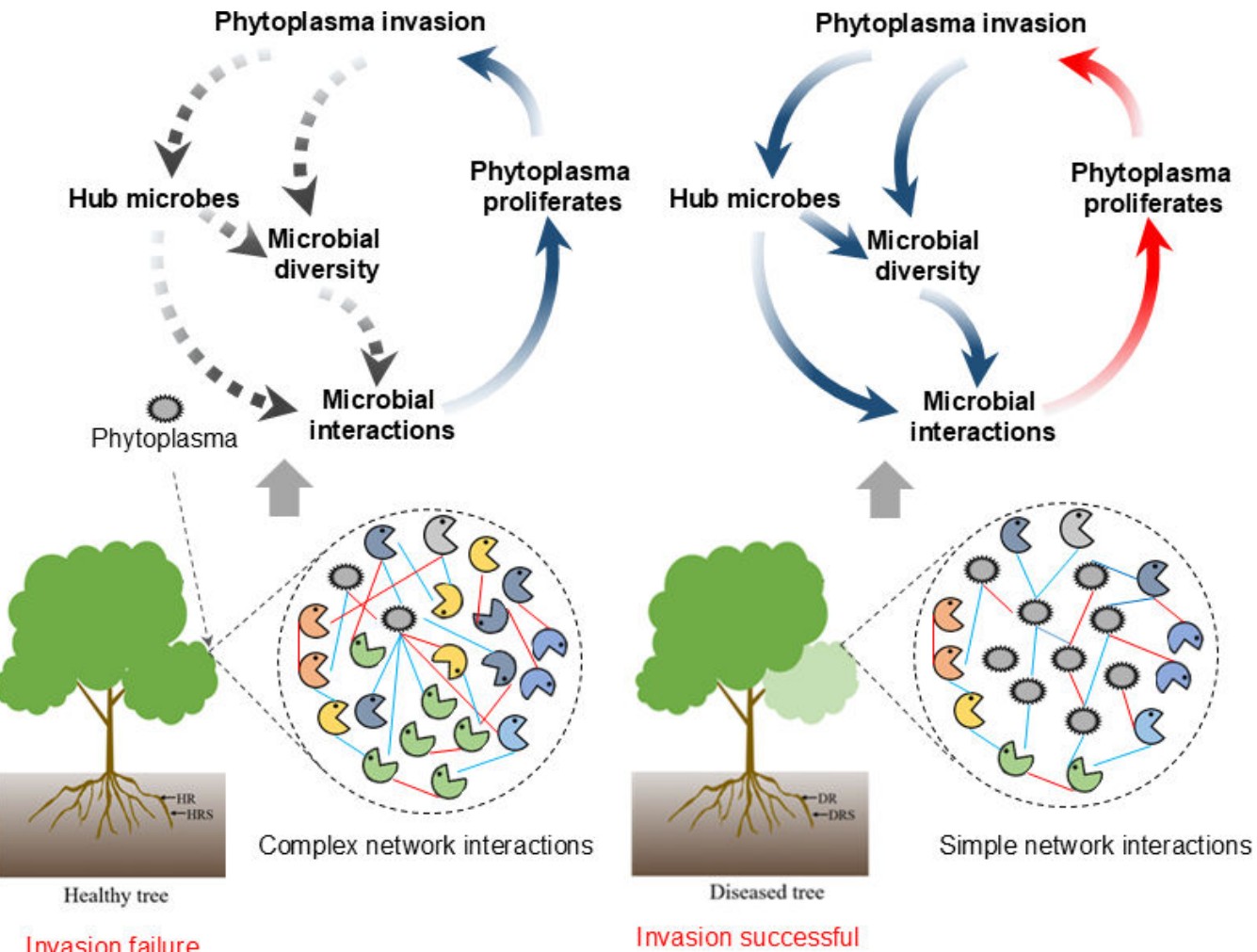

**FIG 9** Relationship between phytoplasma invasion and PaWB. HR: healthy root; HRS: healthy rhizosphere soil; DR: diseased root; DRS: diseased rhizosphere soil.

## Taxonomic restructuring reveals pathogen-induced exclusion of plant-beneficial taxa

At the taxonomic level, PaWB was associated with a pronounced shift from Proteobacteria-dominated communities in healthy tissues to Firmicutes-enriched profiles in symptomatic tissues. Notably, the depletion of well-characterized beneficial genera such as *Sphingomonas* and *Methylobacterium*—likely driven by competitive exclusion or niche disruption—suggests that phytoplasma undermines core members of the beneficial plant microbiota. This observation mirrors findings in poplar (43) and supports the broader hypothesis that pathogen invasion often entails the displacement of cooperative microbial taxa essential for plant homeostasis (2).

Intriguingly, asymptomatic tissues exhibited transient enrichment of these beneficial genera, implying that *Paulownia* may recruit microbial allies during early infection stages—consistent with the "cry-for-help" model observed in *Capsicum*, wheat, and *Arabidopsis* (4–7). Such recruitment may represent a plant-intrinsic microbiome-mediated defense mechanism that delays or suppresses symptom expression.

## Functional reprogramming of endophytes links microbial activity to host physiology

Phytoplasma infection triggers profound functional shifts in the endophytic microbiome, particularly in symptomatic tissues. Key metabolic pathways—including carbohydrate

degradation and tryptophan metabolism—were significantly altered, reflecting a dual mechanism of host manipulation: phytoplasma-induced phloem blockage (via callose deposition (44) disrupts sucrose transport, leading to apoplastic sugar accumulation. This high-carbon microenvironment favors opportunistic microbes (e.g., *Bacillus* and *Pseudomonas*), mirroring observations in Huanglongbing and rice stripe disease (16, 44). Disrupted tryptophan metabolism likely impairs auxin (IAA) biosynthesis (45), both directly and through microbiome-mediated pathways. The resulting hormonal imbalance directly contributes to apical dominance loss and excessive branching, the hallmark symptoms of PaWB (46). Critically, these functional changes suggest that the restructured microbiome is not merely a passive consequence of infection but is very likely an active participant in symptom exacerbation through nutrient niche modification and host developmental interference.

## Implications for early diagnosis and microbiome-informed disease management

The identification of 11 key microbial genera—including *Candidatus* Phytoplasma, *Sphingomonas*, *Pseudomonas*, and *Novosphingobium*—as robust predictors of PaWB status provides a foundation for biomarker-based early diagnostics. In particular, the steep gradient of *Candidatus* Phytoplasma abundance across health states, and its strong association with microbiome collapse, offers a powerful tool for disease monitoring and risk assessment. Furthermore, the observed functional shifts suggest that microbiome manipulation, either through inoculation with beneficial taxa or modulation of community structure, could serve as a novel strategy for PaWB mitigation.

## Limitations and future directions

While our study establishes key correlations between phytoplasma infection, microbiome destabilization, and symptom development, some limitations should be noted. Low-abundance phytoplasma cells detected in asymptomatic tissues may have pre-conditioned the so-called "healthy" microbiome, attenuating observed differences. Moreover, conventional PCR lacks the sensitivity to quantify subtle variations in phytoplasma load or detect early-stage microbial shifts. To address these limitations, future work should adopt absolute quantification approaches (e.g., qPCR) and longitudinal sampling designs stratified by pathogen load to unravel the causality between phytoplasma dynamics and microbiome disruption. Implement absolute quantification methods (e.g., qPCR/digital PCR) to precisely measure phytoplasma titers. Although our study included only 24 samples, the results were robust and statistically significant. To provide a thorough mechanistic understanding, future research will employ longitudinal sampling across infection stages and larger, geographically diverse cohorts to validate these patterns.

## Conclusion

Together, our study provides new mechanistic insights into the pathogenesis of PaWB, revealing that phytoplasma invasion disrupts microbial diversity, interaction networks, and functional capacity in a tissue-specific and symptom-linked manner. These alterations not only reflect but likely facilitate disease progression. By uncovering microbial and metabolic markers of early infection, our work establishes a conceptual framework for microbiome-mediated disease surveillance and paves the way for diagnostic and biocontrol strategies based on microbial ecology.

## ACKNOWLEDGMENTS

We would like to express our gratitude to Dr. Ziwen Yang from South China Normal University for the suggestions he provided during the writing of the paper.

This work was supported by the Henan Postdoctoral Science Foundation (HN2024111), Key Research Project Funding Program of Higher Education Institutions in Henan Province (26A220004), and the Zhongyuan Scholars of Henan Province (30601986). We gratefully acknowledge the financial support provided by these funding agencies, which was instrumental in the successful completion of this study.

## AUTHOR AFFILIATIONS

[1]College of Forestry, Henan Agricultural University, Zhengzhou, China
[2]School of Ecology, Sun Yat-sen University, Shenzhen, China
[3]Henan Academy of Forestry, Zhengzhou, China

## AUTHOR ORCIDs

Xuefei Tang  http://orcid.org/0009-0005-1978-6327
Tuoyan Chen  http://orcid.org/0009-0007-2464-7094
Xuanzhen Li  http://orcid.org/0000-0002-2493-3927
Guoqiang Fan  http://orcid.org/0000-0001-8018-8143

## AUTHOR CONTRIBUTIONS

Xuefei Tang, Conceptualization, Methodology, Resources, Software, Supervision, Validation, Writing – original draft, Writing – review and editing | Tuoyan Chen, Software, Validation, Visualization, Writing – original draft, Writing – review and editing | Xiaoqiao Zhai, Data curation, Funding acquisition, Resources, Software | Jing Huang, Data curation, Formal analysis, Investigation, Methodology, Resources, Software, Validation | Yifei Sun, Software, Validation, Visualization, Writing – review and editing | Yuchen Yang, Resources, Software, Validation, Writing – review and editing | Zhenli Zhao, Formal analysis, Investigation, Methodology, Validation, Visualization | Yanshuo Pan, Data curation, Software, Validation, Visualization | Yuhao Huang, Software, Validation, Visualization | Xuanzhen Li, Conceptualization, Project administration, Supervision | Guoqiang Fan, Conceptualization, Project administration, Supervision, Writing – review and editing

## DATA AVAILABILITY

The original sequencing data have been deposited in the NCBI Sequence Read Archive under the accession number PRJNA821669.

## ADDITIONAL FILES

The following material is available online.

### Supplemental Material

**Fig. S1 (Spectrum01489-25-s0001.tif).** Abundances of Phytoplasma in the different compartments of *Paulownia*.
**Fig. S2 (Spectrum01489-25-s0002.tif).** Abundance comparison of biomarkers in congeneric samples (reads).
**Fig. S3 (Spectrum01489-25-s0003.tif).** Spearman correlation analysis between biomarkers and bacteria α diversity.
**Fig. S4 (Spectrum01489-25-s0004.tif).** Bacterial network interactions in samples with different symptoms.
**Fig. S5 (Spectrum01489-25-s0005.tif).** Linear fitting analysis of bacterial community α diversity and network complexity.
**Fig. S6 (Spectrum01489-25-s0006.tif).** Spearman correlation analysis between biomarkers and network topological parameters in different samples.
**Fig. S7 (Spectrum01489-25-s0007.tif).** PICRUSt2 analysis of metabolic pathways of key metabolic pathway gene abundances in diseased vs. healthy tissues in *Paulownia*.

**Fig. S8 (Spectrum01489-25-s0008.tif).** Comparison of the number of edges and nodes in bacterial community networks among different samples.

**Supplemental material (Spectrum01489-25-s0009.docx).** Supplemental figure captions.

**Supplemental tables (Spectrum01489-25-s0010.docx).** Tables S1 to S5.

## Open Peer Review

**PEER REVIEW HISTORY (review-history.pdf).** An accounting of the reviewer comments and feedback.

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
