## [Reviewer comments · Microbiology Spectrum]

Microbiology Spectrum

Phytoplasma-Induced Alterations in Endophytic Bacterial Communities in *Paulownia*: Implications for Witches' Broom

Xuefei Tang, Tuoyan Chen, Xiaoqiao Zhai, Jing Huang, Yi-Fei Sun, Yuchen Yang, Zhenli Zhao, Yanhuo Pan, Yu-Hao Huang, Xuanzhen Li, and Guoqiang Fan

Corresponding Author(s): Guoqiang Fan, Henan Agricultural University

Review Timeline:

Submission Date:	May 13, 2025
Editorial Decision:	July 10, 2025
Revision Received:	August 11, 2025
Accepted:	August 14, 2025

Editor: Lindsey Burbank

Reviewer(s): Disclosure of reviewer identity is with reference to reviewer comments included in decision letter(s). The following individuals involved in review of your submission have agreed to reveal their identity: Franco Daniel Franco (Reviewer #1)

Transaction Report:

DOI: <https://doi.org/10.1128/spectrum.01489-25>

Re: Spectrum01489-25 (**Phytoplasma-Induced Alterations in Endophytic Bacterial Communities in *Paulownia*: Implications for Witches' Broom**)

Dear Prof. Guoqiang Fan:

Thank you for the privilege of reviewing your work. Below you will find my comments, instructions from the Spectrum editorial office, and the reviewer comments.

Overall, this manuscript contains some interesting findings regarding phytoplasma infection. However, the reviewers raised several points that should be addressed. Note that novelty and hypothesis-driven research questions are not requirements for publication in *Microbiology Spectrum*, and descriptive studies are within our journal scope. In responding to the reviewers comments, please give attention to the methodological questions that were raised, and also the suggestions for added context.

Revision Guidelines

Sincerely,
Lindsey Burbank
Editor
Microbiology Spectrum

Reviewer #1 (Comments for the Author):

This study provides valuable insights into the phytoplasma-induced shifts in endophytic bacterial communities in *Paulownia* and

their implications for Witches' Broom disease. The work is well-structured and employs robust methodologies, but several areas require clarification or improvement to strengthen the manuscript's impact and reproducibility

- 1-The identification of Candidatus Phytoplasma based on microbiome 16S data is too general. Specific PCR and phylogenetic analysis are needed to confirm the phytoplasma strain and determine its 16Sr group/subgroup for proper taxonomic resolution.
- 2-While the manuscript presents functional predictions (e.g., carbohydrate and tryptophan metabolism shifts) based on PICRUSt2, the discussion lacks integration with plant physiological responses. These pathways should be interpreted in the context of Paulownia's defense, stress, or metabolic regulation. Without such links, the functional implications remain speculative. Strengthening the discussion with relevant literature or mechanistic hypotheses would greatly enhance the impact of these findings.
- 3-The manuscript compares bacterial communities across plant compartments (endosphere vs. rhizosphere) but uses different primer sets (V5-V7 vs. V3-V4). This introduces potential amplification bias that can affect observed diversity and composition. The authors should acknowledge this methodological limitation and interpret cross-compartment comparisons with appropriate caution.
- 4-The manuscript should clarify the criteria used to classify asymptomatic samples, particularly whether molecular methods (e.g., qPCR) were employed to confirm phytoplasma levels below a symptomatic threshold. Given the detection of phytoplasma in these tissues, this distinction is critical for interpreting microbial community differences between asymptomatic and healthy samples. Please specify the diagnostic approach and address how potential latent infections might impact the study's conclusions regarding microbiome dynamics

Reviewer #3 (Comments for the Author):

This manuscript explores the mechanisms underlying the impact of Paulownia Witches' Broom (WB) on plant endophytic bacterial communities. The study employed high-throughput 16S rRNA sequencing to analyze differences in microbial composition across various tissue sites (leaves, branches, roots, and rhizosphere soil) between healthy and infected plants. In infected leaves and branches, the relative abundance of phytoplasma was significantly higher than in healthy tissues, while bacterial diversity indices and microbial interaction network complexity decreased in these compartments. Despite notable changes in the above-ground microbial communities, the bacterial composition in roots and rhizosphere soil remained relatively stable. The study further identified 11 key bacterial taxa as disease prediction markers. Functional pathway prediction analysis revealed significant enrichment of pathways related to carbohydrate and amino acid metabolism in infected tissues, suggesting adaptive functional restructuring of the microbial community.

This manuscript has yielded some interesting findings and provides a relatively detailed account of the effects of WB on the microbiota of different Paulownia organs. However, there may be several issues regarding the overall conceptual framework, presentation of research results, and writing style of the manuscript.

Major comments

1. The primary concern is the manuscript's potential absence of sufficient novelty or a well-defined scientific question to drive the research, which may give the impression of an incomplete study.

Although the authors utilized a series of popular microbial ecology analytical methods, the results presented seem primarily descriptive rather than mechanistic. For instance, the authors mainly list various impacts of phytoplasma infection on the microbiome in different plant organs of Paulownia trees (e.g., lines 68-69), but may not have focused on specific or more meaningful scientific questions. I am unsure whether this approach aligns with the journal's expectations for manuscripts.

When plant bacterial communities are invaded by a pathogenic bacterium, the pathogen's substantial proliferation and dominance within the plant bacterial community (e.g., Fig. 5D&E) inevitably alter various ecological characteristics related to alpha- and beta-diversity. Similar studies have been reported frequently; however, based on the presentation of results and the overall discussion in this manuscript, the mechanistic aspects behind these phenomena may not have been thoroughly explored at the levels of plant pathology or microbial ecology. This deficiency is particularly evident in the Discussion section.

Furthermore, the issue of "lacking specific scientific questions" may also be partially reflected in the Introduction. In its current version, the authors do not comprehensively review the specific scientific questions of interest (especially regarding the plant microbiomes) in the Introduction, nor do they raise high-significance scientific questions against the backdrop of relevant literature.

2. Some scientific findings may have been reported previously.

Similar to the point raised in comment 1, some findings in this study may not be considered highly "novel". For example:

(1) The greater impact of phytoplasma on the above-ground (leaves and branches) microbiome compared to the below-ground (roots and rhizosphere) microbiome: Similar trends have been observed in plant microbiomes under various natural (e.g., wildfires, diseases) and anthropogenic disturbances (e.g., mowing, fertilization).

(2) Reduced bacterial diversity under phytoplasma influence: This is likely a secondary effect of the bacterial community being dominated by specific pathogenic taxa (e.g., *Candidatus Phytoplasma*).

(3) Detection of phytoplasma in asymptomatic host plants: Although I am not an expert in phytoplasma, it is well-documented that many pathogenic bacteria/fungi can be widely detected in plants without causing disease, which is a relatively common phenomenon.

This is not to say that such similar findings are unnecessary to mention. However, their superficial treatment in the manuscript limits their scholarly value.

3. The sample size used in the study is relatively small

Overall, this study utilized only 24 samples, including leaves, branches, roots, and rhizosphere soil from six trees (three healthy and three infected). Perhaps collecting samples from slightly more sampling locations or time points could yield more representative or convincing analytical results.

4. There are doubts about the primer selection

The authors used different primers for amplifying and sequencing bacterial communities from plant endophytes (799F/1193R, used for 18 samples) and rhizosphere soil (338F/806R, used for 6 samples). Since different primers have varying biases toward different microbial groups, this may affect the comparability of bacterial community compositions obtained from these different samples (see DOI: <https://doi.org/10.1002/amt.2.135>). Although the 799F/1193R primer pair is commonly used to reduce host sequence contamination and is thus frequently employed for endophytic bacteria, this does not mean it cannot be effectively used for rhizosphere soil studies. Especially for a relatively small-scale study, the primer pair used to obtain marker genes should at least be consistent.

Specific comments

Line 59: "the *Candidatus* genus". Please review this expression, as "*Candidatus*" is not a microbial genus.

Line 62: The capitalization of "Phytoplasma" is inconsistent throughout the manuscript.

Lines 78-80: "Our findings provide...", is this phrasing appropriate for the Introduction?

Line 217: Is it necessary to include references in the Results section?

Line 218: Fig. 5E is mentioned before Figs. 5B-D in the main text. Please check if this is appropriate.

Line 235: Please correct the position of the period.

Line 237: Why is this text in bold?

Lines 238-239 & Line 300: The taxonomic status of *Candidatus Phytoplasma* obtained in this study is not clearly stated in the manuscript. Based on the results, it appears to belong to the phylum Firmicutes. Under this scenario, is it meaningful to solely explore the changes in the abundance of Firmicutes at the phylum level?

Throughout the manuscript: It is recommended to use either first-line indentation or add blank lines between paragraphs in the submitted document (if permitted by the journal).

This study provides valuable insights into the phytoplasma-induced shifts in endophytic bacterial communities in Paulownia and their implications for Witches' Broom disease. The work is well-structured and employs robust methodologies, but several areas require clarification or improvement to strengthen the manuscript's impact and reproducibility

1-The identification of Candidatus Phytoplasma based on microbiome 16S data is too general. Specific PCR and phylogenetic analysis are needed to confirm the phytoplasma strain and determine its 16Sr group/subgroup for proper taxonomic resolution.

2-While the manuscript presents functional predictions (e.g., carbohydrate and tryptophan metabolism shifts) based on PICRUSt2, the discussion lacks integration with plant physiological responses. These pathways should be interpreted in the context of Paulownia's defense, stress, or metabolic regulation. Without such links, the functional implications remain speculative. Strengthening the discussion with relevant literature or mechanistic hypotheses would greatly enhance the impact of these findings.

3-The manuscript compares bacterial communities across plant compartments (endosphere vs. rhizosphere) but uses different primer sets (V5-V7 vs. V3-V4). This introduces potential amplification bias that can affect observed diversity and composition. The authors should acknowledge this methodological limitation and interpret cross-compartment comparisons with appropriate caution.

4-The manuscript should clarify the criteria used to classify asymptomatic samples, particularly whether molecular methods (e.g., qPCR) were employed to confirm phytoplasma levels below a symptomatic threshold. Given the detection of phytoplasma in these tissues, this distinction is critical for interpreting microbial community differences between asymptomatic and healthy samples. Please specify the diagnostic approach and address how potential latent infections might impact the study's conclusions regarding microbiome dynamics

Dear Editor,

Thank you very much for handling and favorable consideration our manuscript entitled “Phytoplasma-Induced Alterations in Endophytic Bacterial Communities in *Paulownia*: Implications for Witches’ Broom” (Spectrum01489-25). We sincerely appreciate the opportunity to publish our work in Microbiology Spectrum. We are grateful for the thoughtful and constructive comments provided by you and the reviewers, which have greatly helped us improve the clarity, rigor, and scientific quality of our manuscript. We have carefully addressed all the suggestions and revised the manuscript accordingly. Please find below our point-by-point responses. The reviewers’ original comments are presented in black, followed by our responses in bluefont.

Sincerely,

Guoqiang Fan

Overall, this manuscript contains some interesting findings regarding phytoplasma infection. However, the reviewers raised several points that should be addressed. Note that novelty and hypothesis-driven research questions are not requirements for publication in Microbiology Spectrum, and descriptive studies are within our journal scope. In responding to the reviewers comments, please give attention to the methodological questions that were raised, and also the suggestions for added context.

Response: We appreciate the constructive suggestions provided by the editor and reviewers. In response, we have rephrased the target sentence and thoroughly revised the manuscript to enhance clarity and precision. Additionally, we added results from nested PCR experiments, scanning electron microscopy to demonstrate the tissue architecture, and an analysis of endophytic network responses to further strengthen the conclusions of the study. These revisions have significantly improved the manuscript. Please find our detailed point-by-point responses below

Response to Reviewer #1

This study provides valuable insights into the phytoplasma-induced shifts in endophytic bacterial communities in *Paulownia* and their implications for Witches’ Broom disease. The work is well-structured and employs robust methodologies, but several areas require clarification or improvement to strengthen the manuscript's impact and reproducibility.

1. “The identification of Candidatus Phytoplasma based on microbiome 16S data is too general. Specific PCR and phylogenetic analysis are needed to confirm the phytoplasma strain and determine its 16Sr group/subgroup for proper taxonomic resolution.”

Response: Thank you for this important comment. We agree that accurate phytoplasma identification requires more specific molecular tools. We added nested PCR experiments to verify the presence of phytoplasma detected by 16S rRNA in symptomatic, asymptomatic, and root tissues (lines 94–104, Methods, Sec. 2.1.2; Fig. 4A–B). The results have confirmed that the power of 16Sr data in identification and classification of phytoplasma groups for the current study.

2. “While the manuscript presents functional predictions (e.g., carbohydrate and tryptophan metabolism shifts) based on PICRUSt2, the discussion lacks integration with plant physiological responses. These pathways should be interpreted in the context of *Paulownia*'s defense, stress, or metabolic regulation. Without such links, the functional implications remain speculative. Strengthening the discussion with relevant literature or mechanistic hypotheses would greatly enhance the impact of these findings.”

Response: We appreciate this insightful suggestion. In the revised Discussion (lines 355–362), we now elaborate on how phytoplasma-induced reduction of key microbial functions (e.g., carbohydrate metabolism and tryptophan biosynthesis) may impair phloem transport and hormone balance in *Paulownia*. These disruptions potentially contribute to symptom development. We also incorporated recent literatures to support the mechanistic link between microbial function and host physiology under phytoplasma stress.

3. “The manuscript compares bacterial communities across plant compartments (endosphere vs. rhizosphere) but uses different primer sets (V5-V7 vs. V3-V4). This introduces potential amplification bias that can affect observed diversity and composition. The authors should acknowledge this methodological limitation and interpret cross-compartment comparisons with appropriate caution.”

Response: Thank you for pointing this out. We apologize for the confusion and the mistake we made. Actually, all our samples were amplified using the 799F/1193R primer pair targeting the V5–V7 region, but in some sentences, we wrongly said we used the V3–V4 primers in the manuscript. We have revised our statements in the Method section (Sec. 2.3, lines 119–128) to make sure all are correctly elaborated that we only used the V5–V7 primers for the entire study.

4. “The manuscript should clarify the criteria used to classify asymptomatic samples, particularly whether molecular methods (e.g., qPCR) were employed to confirm phytoplasma levels below a symptomatic threshold. Given the detection of phytoplasma in these tissues, this distinction is critical for interpreting microbial community differences between asymptomatic and healthy samples. Please specify the diagnostic approach and address how potential latent infections might impact the study's conclusions regarding microbiome dynamics”

Response: We appreciate this valuable suggestion. As clarified in the Methods (Line 109-110), asymptomatic samples were defined by the absence of visible PaWB symptoms in diseased *Paulownia*. We further verified phytoplasma presence using nested PCR and found faint bands in asymptomatic samples, indicating a low phytoplasma load (lines 175-178; Fig 4 A, B). Although we agree that qPCR would offer better quantification, we were unable to perform it due to sample constraints. This limitation is now discussed in the Discussion (lines 368–375), and future work will incorporate qPCR to better define infection thresholds.

Response to Reviewer #3

This manuscript explores the mechanisms underlying the impact of Paulownia Witches' Broom (WB) on plant endophytic bacterial communities. The study employed high-throughput 16S rRNA sequencing to analyze differences in microbial composition across various tissue sites (leaves, branches, roots, and rhizosphere soil) between healthy and infected plants. In infected leaves and branches, the relative abundance of phytoplasma was significantly higher than in healthy tissues, while bacterial diversity indices and microbial interaction network complexity decreased in these compartments. Despite notable changes in the above-ground microbial communities, the bacterial composition in roots and rhizosphere soil remained relatively stable. The study further identified 11 key bacterial taxa as disease prediction markers. Functional pathway prediction analysis revealed significant enrichment of pathways related to carbohydrate and amino acid metabolism in infected tissues, suggesting adaptive functional restructuring of the microbial community.

This manuscript has yielded some interesting findings and provides a relatively detailed account of the effects of WB on the microbiota of different *Paulownia* organs. However, there may be several issues regarding the overall conceptual framework, presentation of research results, and writing style of the manuscript.

Major comments

1. The primary concern is the manuscript's potential absence of sufficient novelty or a well-defined scientific question to drive the research, which may give the impression of an incomplete study.

Although the authors utilized a series of popular microbial ecology analytical methods, the results presented seem primarily descriptive rather than mechanistic. For instance, the authors mainly list various impacts of phytoplasma infection on the microbiome in different plant organs of *Paulownia* trees (e.g., lines 68-69), but may not have focused on specific or more meaningful scientific questions. I am unsure whether this approach aligns with the journal's expectations for manuscripts.

When plant bacterial communities are invaded by a pathogenic bacterium, the pathogen's substantial proliferation and dominance within the plant bacterial community (e.g., Fig. 5D&E) inevitably alter various ecological characteristics related to alpha- and beta-diversity. Similar studies have been reported frequently; however, based on the presentation of results and the overall discussion in this manuscript, the mechanistic aspects behind these phenomena may not have been thoroughly explored at the levels of plant pathology or microbial ecology. This deficiency is particularly evident in the Discussion section.

Furthermore, the issue of "lacking specific scientific questions" may also be partially reflected in the Introduction. In its current version, the authors do not comprehensively review the specific scientific questions of interest (especially regarding the plant microbiomes) in the Introduction, nor do they raise high-significance scientific questions against the backdrop of relevant literature.

Response: Thank you for this important comment. In this study, we performed an integration analysis of high-throughput amplicon sequencing, random Forest analysis and co-occurrence network analysis to investigate the impact of witches' broom disease on endophytic microbiome in *Paulownia* plants. Our results showed that phytoplasma infection reshapes the endophytic microbiome in a compartment-specific manner and their composition, abundance and interaction strength would be changed with the prolongation of symptom development. Although it is well known that pathogen invasion can alter plant endophytic microbiomes, no previous study has addressed these effects in *P. fortunei*. To our knowledge, this work is the first to characterize how phytoplasma infection impacts endophytic community composition, richness, and co-occurrence relationships in *P. fortunei*. Furthermore, Random Forest analysis identified 11 bacterial genera—*Sphingomonas*, *Pseudomonas*, *Bacillus*, *Methylobacterium*, *Enterobacter*, *Rhizobium*, *Curtobacterium*, *Microbacterium*, *Pantoea*, *Novosphingobium*, and *Chryseobacterium*. These taxa may influence host nutrient fluxes and defense signaling under infection. Together, our findings provide novel insights into host–microbe interactions during WB disease and establish a foundation for microbiome-based management strategies. We emphasized mechanistic interpretations and their significance across multiple sections (e.g., Introduction, lines 48-56, Discussion, lines 294-297).

2. Some scientific findings may have been reported previously.

Similar to the point raised in comment 1, some findings in this study may not be considered highly "novel". For example:

(1) The greater impact of phytoplasma on the above-ground (leaves and branches) microbiome compared to the below-ground (roots and rhizosphere) microbiome: Similar trends have been observed in plant microbiomes under various natural (e.g., wildfires, diseases) and anthropogenic disturbances (e.g., mowing, fertilization).

(2) Reduced bacterial diversity under phytoplasma influence: This is likely a secondary effect of the bacterial community being dominated by specific pathogenic taxa (e.g., *Candidatus Phytoplasma*).

(3) Detection of phytoplasma in asymptomatic host plants: Although I am not an expert in phytoplasma, it is well-documented that many pathogenic bacteria/fungi can be widely detected in plants without causing disease, which is a relatively common phenomenon.

This is not to say that such similar findings are unnecessary to mention. However, their superficial treatment in the manuscript limits their scholarly value.

Response: Thank you for this important comment. We acknowledge that similar trends have been observed in other host–pathogen systems. Our study is, to our knowledge, the first to demonstrate that phytoplasma infection precipitates a tissue-specific collapse of the *Paulownia* endophytic microbiome—disrupting both network structure and functional capacity well before any visual symptoms emerge. Moreover, by applying a Random Forest classifier, we uncovered 11 bacterial genera predictive of PaWB status—several of which have never before been linked to this disease. We have now underscored these two novelty advances both in the Abstract (lines 27–31) and in the Discussion (lines 323–328).

3. The sample size used in the study is relatively small

Overall, this study utilized only 24 samples, including leaves, branches, roots, and rhizosphere soil from six trees (three healthy and three infected). Perhaps collecting samples from slightly more sampling locations or time points could yield more representative or convincing analytical results.

Response: We appreciate the reviewer's concern. We have now acknowledged this limitation in the Discussion (lines 368-372) and emphasized that, despite the modest sample size, the findings were consistent and statistically supported. Future studies will expand sampling across multiple seasons and geographic regions to provide a thorough mechanistic understanding.

4. There are doubts about the primer selection

The authors used different primers for amplifying and sequencing bacterial communities from plant endophytes (799F/1193R, used for 18 samples) and rhizosphere soil (338F/806R, used for 6 samples). Since different primers have varying biases toward different microbial groups, this may affect the comparability of bacterial community compositions obtained from these different samples (see DOI: <https://doi.org/10.1002/imt2.135>). Although the 799F/1193R primer pair is commonly used to reduce host sequence contamination and is thus frequently employed for endophytic bacteria, this does not mean it cannot be effectively used for rhizosphere soil studies. Especially for a relatively small-scale study, the primer pair used to obtain marker genes should at least be consistent.

Response: Thank you for pointing this out. We apologize for the confusion and the mistake we made. Actually, all our samples were amplified using the 799F/1193R primer pair targeting the V5–V7 region, but in some sentences, we wrongly said we used the V3–V4 primers in the manuscript. We have revised our statements in the Method section (Sec. 2.3, lines 124–128) to make sure all are correctly elaborated that we only used the V5–V7 primers for the entire study.

Specific Comments:

Line 59: "the Candidatus genus". Please review this expression, as "Candidatus" is not a microbial genus.

Response: Thank you for this important comment. We have thoroughly reviewed the entire manuscript and corrected to *Candidatus* phytoplasma.

Line 62: The capitalization of "Phytoplasma" is inconsistent throughout the manuscript.

Response: Thank you for this important comment. We have made the necessary formatting revisions and standardized capitalization of "Phytoplasma". Only the first letter of each sentence is capitalized, with the remaining text in lowercase as appropriate.

Lines 78-80: "Our findings provide...", is this phrasing appropriate for the Introduction?

Response: Thank you for this important comment. We have removed such expressions to ensure the content is appropriate for the Introduction section.

Line 217: Is it necessary to include references in the Results section?

Response: Thank you for this important comment. We have removed the reference in the Results section.

Line 218: Fig. 5E is mentioned before Figs. 5B-D in the main text. Please check if this is appropriate.

Response: Thank you for this important comment. We have made modifications and adjustments.

Line 235: Please correct the position of the period.

Response: Thank you for this important comment. We have thoroughly reviewed the entire manuscript and corrected.

Line 237: Why is this text in bold?

Response: Thank you for this important comment. We have removed unintended bold.

Lines 238-239 & Line 300: The taxonomic status of Candidatus Phytoplasma obtained in this study is not clearly stated in the manuscript. Based on the results, it appears to belong to the phylum Firmicutes. Under this scenario, is it meaningful to solely explore the changes in the abundance of Firmicutes at the phylum level?

Response: Thank you for this important comment. We have added explanation that Firmicutes dominance reflects phytoplasma proliferation, not general phylum shifts and subsequent analyses therefore focused on genus-level resolution (lines 244-247).

Throughout the manuscript: It is recommended to use either first-line indentation or add blank lines between paragraphs in the submitted document (if permitted by the journal).

Response: Thank you for this important suggestion. In the submitted document, we have used consistent first-line indentation and removed paragraph spacing in line with Microbiology Spectrum guidelines.

Re: Spectrum01489-25R1 (**Phytoplasma-Induced Alterations in Endophytic Bacterial Communities in *Paulownia*: Implications for Witches' Broom**)

Dear Prof. Guoqiang Fan:

Your manuscript has been accepted, and I am forwarding it to the ASM production staff for publication. Your paper will first be checked to make sure all elements meet the technical requirements. ASM staff will contact you if anything needs to be revised before copyediting and production can begin. Otherwise, you will be notified when your proofs are ready to be viewed.

Sincerely,
Lindsey Burbank
Editor
Microbiology Spectrum